# Wearable Cardiopulmonary Function Evaluation System for Six-Minute Walking Test

**DOI:** 10.3390/s19214656

**Published:** 2019-10-26

**Authors:** Bor-Shing Lin, Ruei-Jie Jhang, Bor-Shyh Lin

**Affiliations:** 1Department of Computer Science and Information Engineering, National Taipei University, New Taipei City 23741, Taiwan; bslin@mail.ntpu.edu.tw; 2Institute of Photonic System, National Chiao Tung University, Tainan 71150, Taiwan; a313255a@gmail.com; 3Institute of Imaging and Biomedical Photonics, National Chiao Tung University, Tainan 71150, Taiwan

**Keywords:** cardiopulmonary function, 6-min walking test (6MWT), electrocardiogram, breathing sound, indoor walking distance

## Abstract

As a submaximal exercise test, a 6-min walking test (6MWT) can be considered a suitable index for the exercise capacity of patients with a respiratory problem. Traditionally, medical staff manually collect cardiopulmonary information using different devices. However, no integrated monitoring system is currently available to simultaneously record the real-time breathing sound, heart rhythm, and precise walking information (i.e., walking distance, speed, and acceleration) during the 6MWT. In this study, a wearable and wireless multiparameter monitoring system is proposed to simultaneously monitor the breathing sound, oxygen saturation (SpO2), electrocardiograph (ECG) signals, and precise walking information during the 6MWT. Here, a wearable mechanical design was successfully used to reduce the effect of motion artifacts on the breathing sound and ECG signal. A multiparameter detection algorithm was designed to effectively estimate heart and breathing rates. Finally, the cardiopulmonary function of smokers was evaluated using the proposed system. The evaluation indicated that this system could reveal dynamic changes and differences in the breathing rate, heart rate, SpO2, walking speed, and acceleration during the 6MWT. The proposed system can serve as a more integrated approach to monitor cardiopulmonary parameters and obtain precise walking information simultaneously during the 6MWT.

## 1. Background

As a submaximal exercise test, the 6-minute walking test (6MWT) can be a suitable index for the exercise capacity of patients with respiratory problems [1,2]. The 6MWT is a simple and inexpensive test that provides the global and integrated response of pulmonary and nonpulmonary factors [3,4]. In the 6MWT, blood pressure, heart rate, and oxygen saturation (SpO2) are monitored before and after the test, and the 6-minute walking distance (6MWD) is recorded as an exercise capacity index [5]. 6MWT provides information of functional capacity, response to therapy, and prognosis across a range of chronic cardiopulmonary conditions. A change in walking distance of more than 50 m is clinically significant in most disease states. A distance less than 350 m is associated with increased mortality in chronic obstructive pulmonary disease, chronic heart failure, and pulmonary arterial hypertension [6].

In clinical applications, the degree of airflow limitation is frequently used to understand the severity of respiratory problems [7,8]. Furthermore, respiratory and limb muscle dysfunction are frequently observed in people with respiratory problems [9,10,11]. The respiratory limitation and peripheral muscle weakness may cause physical deterioration and reduce exercise ability [12,13,14]. Previous studies have indicated that the exercise capacity of patients with respiratory problems may be a better predictor of mortality than the forced expiratory volume in one second (FEV1), body mass index (BMI), and associated comorbidities [15]. Therefore, the 6MWT is the most commonly used test to evaluate exercise capacity and predict functional outcomes in patients with respiratory problems [16,17,18].

Some people with cardiopulmonary problems can develop abnormal breathing or arrhythmia during the 6MWT, which can remain undetected without real-time breathing sound and heart rhythm monitoring. Therefore, developing a system that provides both real-time breathing sound and heart rhythm monitoring is essential.

## 2. Related Work

Several monitoring systems have been proposed to monitor the exercise capacity and bioparameters of patients during the 6MWT. In 2009, Jehn et al. monitored the step count and three-dimensional signal vector magnitude (SVM) of patients with chronic heart failure by using an accelerometer and a pedometer during the 6MWT, and they attempted to obtain alternative performance parameters gained from accelerometers related to 6MWD [19]. They showed that the step frequency and sum of SVM may only be correlated with the high-intensity 6MWD. In 2011, Charles et al. proposed a remote monitoring system to monitor electrocardiograph (ECG) signals and gait information from cardiac patients during the 6MWT [20]. A miniaturized heart and activity monitor placed on a belt was used to record single-lead ECG signals, and a nondifferential global positioning system (GPS) receiver was used to record walking speed, elapsed distance, and location. However, the GPS technique is impractical for an indoor 6MWT. In 2015, Juen et al. used machine learning techniques on a smartphone to predict the walking speed and distance of patients during the 6MWT and natural free walking [21]. A smartphone accelerometer provided three-axial acceleration, which were used to estimate gait speed and predict the 6MWD. However, this technique requires pretraining and may vary for different individuals. In 2015, Nicole et al. proposed a calibration-free algorithm for the 6MWT to record the total walking distance, step timing, and walking changes over time from information obtained using the accelerometer gyroscope in the smartphone [22]. Gait was estimated using the number of foot strikes obtained from the smartphone and a known walkway length. However, larger distance error may occur in some conditions, including when the trial ended shortly after a turn, in populations that do not have distinct foot strikes (i.e., shuffle gait with stroke or elderly), and those with severe gait asymmetry. In 2016, Ruth et al. designed a remote monitoring system to monitor the heart rate and ECG signals of older adults with heart failure during the 6MWT [23]. Moreover, two electrodes were placed on the back of an iPhone to measure real-time, single-lead ECG signals, whereas the accelerometer and GPS in the iPhone were used to collect 6MWD data. However, a user was required to hold this device in their hand; therefore, this technique may be unsuitable for the 6MWT. The aforementioned 6MWT system could not provide real-time breathing sound and heart rhythm monitoring to observe the occurrences of abnormal breathing activities and cardiac arrhythmia during the 6MWT and avoid test risks. Moreover, they could not provide precise walking information.

Several physiological monitoring systems have also been developed to monitor cardiopulmonary parameters. In 2014, Andreoni et al. designed a wearable monitoring device to monitor the physiological parameters of patients with postpolio syndrome during the 6MWT to evaluate the functional efficiency of their lower limb orthosis [24]. A commercial ECG, impedance-cardiograph (ICG) signal acquisition model, and triaxial accelerometer were integrated into this system, and the system could be placed on the lumbar area of the user by using an elastic belt. However, the influence of motion artifacts on the measurement of ECG and ICG was large. Moreover, gait information estimated using the walking step and step length was inaccurate. In 2017, Miramontes et al. proposed a mobile healthcare platform (PlaIMoS) to monitor the cardiovascular and respiratory parameters of patients [25]. This system comprises a wearable device, measurement station, and wireless sensor network (WSN) infrastructure. The aforementioned wearable device was integrated with a three-dimensional accelerator, ECG sensor, and temperature sensor. The information of vertical or horizontal positions obtained from the three-dimensional accelerator could be used to estimate the state of falling or standing. The aforementioned measurement station integrated a photoplethysmography (PPG) sensor, an airflow sensor, and a galvanic sensor to monitor the SpO2 level, breathing rate, and skin response voltage of the user. The obtained physiological parameters were then transmitted to the nearest communication node of the WSN. However, the cost of the WSN infrastructure increased with the increasing WSN coverage. Moreover, only ECG and body temperature could be monitored under motion. In 2018, Taffoni et al. proposed a wearable system for the continuous and real-time monitoring of walking steps and cardiovascular and respiratory parameters to help people promote their health and exercise adherence [26]. A commercial headphone with a microphone stick, a commercial PPG sensor, an inertial-magnetic unit sensor, and a differential pressure sensor were integrated into the aforementioned wearable system. Information was obtained using the triaxial accelerometer signal, heart rate was estimated using the PPG signal, and breathing rate was estimated from the signal of exhalation and inhalation obtained by the differential pressure sensor. However, the accuracy of heart rate estimation from the PPG signal was unsatisfactory, particularly under motion. Moreover, placing the differential pressure sensor near the nose easily was uncomfortable and inconvenient. In 2019, Klum et al. proposed a wearable multimodal stethoscope patch for wireless biosignal acquisition and long-term auscultation [27]. The authors presented the prototype of a wearable, Bluetooth 5.0 LE-enabled multimodal sensor patch combining five modalities: Microelectromechanical systems (MEMS) stethoscope, ambient noise sensing, ECG, impedance neumography, and nine-axial actigraphy. However, this system is mainly applied in sleep monitoring and general postoperative care, not in 6MWT. It also lacks walking information acquisition and interference (caused by motion) removal.

## 3. Motivation

To overcome the aforementioned issue, a wearable wireless cardiopulmonary function evaluation system is proposed in this study. In this study, a wireless wearable multiparameter acquisition device was developed to simultaneously monitor the real-time breathing sound, ECG signals, and SpO2 under motion. Moreover, by using the wearable mechanical design, the device could successfully reduce the influence of motion on monitoring breathing sounds and ECG signals. To avoid differences between the walking distance of different individuals in the aforementioned methods, radio frequency identification (RFID) was employed for the measurement of the indoor 6MWD. A multiparameter analysis algorithm was designed to automatically extract heart rate, breathing rate, walking distance, and speed. Finally, the proposed system was validated, and cardiopulmonary functions of smoking and nonsmoking groups under the 6MWT were investigated. Differences in the breathing rate, walking distance, and speed between smoking and nonsmoking groups during the 6MWT were found to be significant.

## 4. System Design and Implementation

### 4.1. System Architecture

Figure 1 shows the scheme of the proposed wearable cardiopulmonary function evaluation system, which primarily included a wireless multiparameter acquisition device, an indoor walking distance measurement device, and a host system. This multiparameter acquisition device was designed to simultaneously monitor multiple signals, such as breathing sound, heart rate, and SpO2, under motion. The acoustic sensor of the multiparameter acquisition device was placed on the upper right anterior chest of the patient to acquire breathing sounds. ECG electrodes were placed on the upper right and left chest as a primary input, and a reference electrode was placed on the lower right of the body close to the right leg to collect the lead I ECG signal. An optical probe was attached on the ear to acquire the PPG signal and then estimate the SpO2 of the user. To reduce the influence of motion, this multiparameter acquisition device used a mechanical design to provide a suitable pressure for maintaining an excellent contacting condition of the device and chest wall. The indoor walking distance measurement device was designed to record and monitor the walking distance, speed, and acceleration. Moreover, the device was tied on the rear heel to monitor walking information. All raw data acquired by the wireless multiparameter acquisition device and indoor walking distance measurement module were simultaneously transmitted to the host system via Bluetooth. A multiparameter monitoring program in the host system was then displayed and analyzed, and these physiological signals were stored to assist the physician in evaluating the cardiopulmonary function of the patient.

### 4.2. Wearable Multiparameter Acquisition Device

The wearable multiparameter acquisition device primarily comprised a wearable mechanical design and wireless biosignal acquisition module. Figure 2 shows the block diagram of the wireless biosignal acquisition module. The model mainly comprises a light-emitting diode (LED) driving circuit, photodiode (PD) amplification circuit, frontend amplifier circuits, a microphone driving circuit, a microprocessor, and a wireless transmission circuit. An external optical probe, which consists of a three-wavelength LED (SMT640/700/910, EPITEX, Japan) and photodiode (PD15-22C/TR8, EVERLIGHT, Taiwan), was used to acquire a PPG signal to estimate SpO2. The power of the LED light source was approximately 0.1 W. In the wireless biosignal acquisition module, the LED driving circuit was designed to provide a steady current for driving a three-wavelength LED of the optical probe, and the PD amplification circuit was designed to amplify the light signal obtained by the photodiode and penetrating through the human tissue. The gain of the PD amplification circuit contains a transimpedance amplifier with the gain of 54 × 104 V/A and a low-pass filter with the cut-off frequency of 64 Hz. Moreover, an external acoustic sensor, which comprises a stethoscope bell (HarveyTM DLX, Welch Allyn, Skaneateles Falls, NY, USA) and microphone (JL-0627C, JEOLUEN, Taiwan), was used to collect breathing sounds. In the wireless biosignal acquisition module, the microphone driving circuit was designed to provide a stable driving voltage for the microphone to avoid the influence of the voltage variation of the power source. The electrical signals of the breathing sound and ECG signals were amplified and filtered using frontend amplifier circuits. The total gains of frontend amplifier circuits were 1000 times and 430 times for the ECG signal and breathing sound, respectively. The frequency bands of frontend amplifier circuits were higher than 0.1 Hz for the ECG to remove the lower frequency interference caused by motion, and were higher than 100 Hz for the breathing sound. After preprocessing these biosignals, they were digitized using a 16-channel 12-bit analog-to-digital converter in the microprocessor (RX210, Renesas, Japan) at a sampling rate of 2048 Hz, and these digital signals were sent to the wireless transmission circuit for transmitting to the host system. The wireless transmission circuit primarily comprised a printed circuit board antenna and Bluetooth module (Ct-BT02, Connectec, Taiwan) with a Bluetooth v2.0+EDR specification. The wearable multiparameter acquisition device operates with a DC power supply of 3.7 V and can continuously operate for more than 6 h with a 1400-mAh Li-ion battery. The size of this module was approximately 60 × 60 × 20 mm^3^.

Figure 3 shows the photograph of the mechanical design in the wearable multiparameter acquisition device. It primarily consisted of a shoulder brace and an elastic band. Velcro on the shoulder brace was used to fix the wireless biosignal acquisition module. Moreover, by adjusting the constriction of the elastic band, the wireless biosignal acquisition module could easily fit the chest contour of the patient to maintain a suitable contracting condition for reducing the effect of motion artifacts.

### 4.3. Indoor Walking Distance Measurement Device

Figure 4 shows the indoor walking distance measurement device. It comprised a 25-m skidproof mat equipped with RFID tags and a wearable RFID reader. RFID tags (RF-QBK05, Guangzhou Dongxin Intellectual Technology CO. LTD, China) were embedded into the back of a skidproof mat, and the distance between every two tags was 9 cm. These RFID tags worked at a radiofrequency of 125 kHz. Each tag comprised a distinct 64-bit ID and could be used to record the patient location to calculate walking information. In this device, an RFID reader (ID-20LA, ID Innovations, Australia) was tied on the rear heel to access data on these RFID tags for recognizing and transmitting the indoor location of the patient to the host system through a 100-m long distance transmission Bluetooth module (DXT3C, DinXing Technology, China). The average error of the measured walking information is 1.3%.

### 4.4. Host System

In this study, a commercial laptop with Microsoft Windows 10 was used as a host platform. A multiparameter monitoring program in the host system was developed in Microsoft C# to monitor real-time ECG signals (heart rate), SpO2, breathing sound signals, and gait information (indoor walking distance, speed, and acceleration).

## 5. System Software Design

The software architecture includes three parts: A graphics user interface (GUI), BUFFER, and THREAD. A part of the GUI provides the ability of precisely displaying and controlling GUI elements. BUFFER is a link-list container and was temporarily used to store received raw data. THREAD denotes the thread of execution, which is the smallest sequence of programmed instructions independently managed using an operating system scheduler. The proposed program comprised three independent threads: A Bluetooth application programming interface (API), Bioanalysis, and NAudio API [28]. The Bluetooth API was used to connect the host system to the wearable multiparameter acquisition device and indoor walking distance measurement device via Bluetooth, and it was used to store raw data into BUFFER. The design of the Bioanalysis thread was based on the proposed multiparameter analysis algorithm for detecting heart rate, breathing rate, SpO2, and gait information. Figure 5 shows the flowchart of the multiparameter monitoring program. First, the GUI allows the user to set or operate this program. The thread of the Bluetooth API was then used to simultaneously search for the wearable multiparameter acquisition device and indoor walking distance measurement device. After the two devices were found, a serial port profile between the host system and two devices was built. The Bluetooth API received raw data obtained from the two devices and stored them into BUFFER. The Bioanalysis thread calculated heart rate, breathing rate, SpO2, and walking distance from ECG signals, breathing sound, PPG signals, and RFID tag information, respectively. Finally, the received breathing sound was sent to the NAudio API to play the real-time breathing sound from the host system.

## 6. Multiparameter Analysis Algorithm

Figure 6a shows the procedure of the proposed multiparameter analysis algorithm. Figure 6b–e show the physiological signals and outputs data after analysis algorithm processing, including the sound signal, ECG signal, fractal dimension (FD) value of the sound signal, PPG signal, breathing events, and R-wave events. To calculate the heart and breathing rates, raw ECG and breathing sound signals were first preprocessed through digital filters. A high-pass filter with a frequency band higher than 100 Hz was applied for the breathing sound signal [29] to remove 60-Hz power line interference and other low-frequency motion interference and to reserve its essential characteristics. A low-pass filter with a frequency band of less than 50 Hz was applied in the ECG signal to eliminate power line interference and high-frequency noise. FD algorithms were used to calculate the FD value of breathing sounds [30]. The FD technique is generally used to estimate the complexity of a geometrical figure and has been widely employed to analyze transient and biomedical signals [31,32,33,34]. The variation in the FD value reflects the instantaneous variation in breathing sounds. Therefore, during a breathing activity, the FD value of breathing sounds increases. According to Katz’s algorithm [35], the FD value of a curve can be given as follows:(1)$$FD=\frac{\log\left(n\right)}{\log\left(n\right)+\log\left(d/L\right)},$$
where *L*, *d*, and *n* denote the total length of the curve, the farthest distance between the first point and any point on the curve, and the step number of the curve, respectively. The step number can be calculated as $n=L\;/\;\bar{a}$, where $\bar{a}$ is the average length between any two nearest discrete points on the curve. A first derivative approach [36] was used to detect the local maximum value of ECG signals and the FD value of breathing sounds. If the local maximum value was higher than the provided dynamic threshold, then it could be considered an R-wave event in ECG or breath activity. The averaged values of ECG or FD value for the first 10 s of breathing sound were defined as dynamic thresholds for detecting R-wave events in ECG and breathing activities. Finally, heart and breathing rates could be estimated from detected R-wave events in ECG and breathing activities.

In this study, the Lambert–Beer law was used to estimate SpO2 [37,38]. Before estimating SpO2, the DC and alternative current (AC) parts in PPG signals must be separated. Every local minimum and local maximum on PPG were first detected. The first derivative approach [39] was used to detect the local extremes of the PPG signal. When the detected local extreme was higher than the provided threshold, it could be considered a local maximum; otherwise, it could be considered a local minimum. The provided threshold was defined as the average of data obtained within the first 0.5 s [40]. After detecting these local minimums and maximums on the PPG, each local minimum can be viewed as the baseline part of PPG, and the difference between the nearest local minimum and maximum can be viewed as the variation part of PPG. The ratio of R can be calculated using the baseline and variation parts of PPG, which can be given as follows:(2)$$R=\frac{\frac{I_{var\left[red\right]}}{I_{bl\left[red\right]}}}{\frac{I_{var\left[nir\right]}}{I_{bl\left[nir\right]}}},$$
(3)SpO2=aR+b.

Here, $I_{var\left[red\right]}$ and $I_{var\left[nir\right]}$ denote variation amplitudes of PPG signals for the red and near-infrared (NIR) light wavelength, respectively, whereas $I_{bl\left[red\right]}$ and $I_{bl\left[nir\right]}$ denote the baseline amplitudes of PPG signals for red and NIR light wavelength, respectively. In the calibration experiment, the participants were instructed to equip the designed device and a commercial pulse oximeter (DB11, DELBio, Taiwan) simultaneously. Next, these participants would be instructed to hold their breath for about 70 s to collect the varying SpO2 data. Finally, a linear regression method was used to obtain coefficients *a* and *b* [41] by using R values obtained from the designed device and SpO2 values obtained from the commercial pulse oximeter.

The received RFID tag information contains a distinct 64-bit ID. If the latest RFID tag information was different from the previous RFID tag information, the walking distance could be immediately calculated using the previously received RFID tag information. Finally, the averaged walking speed could be estimated from the elapsed time between two locations that was tagged off a nearest RFID tag every 20 s.

## 7. Experiment Design and Procedures

In this study, the institutional review board (IRB 103-3295A3) approved the clinical experiment, Chang Gung Medical Foundation, Taiwan, and the informed consent was signed. The 6MWT was performed at National Chiao Tong University, Taiwan. Participants were consecutively monitored for physiological signals by using conventional devices, namely a pulse oximeter (SB100, Rossmax, Taiwan), a blood pressure monitor (ES-P370, Terumo, Japan), and the proposed system during the 6MWT. Each participant was monitored using the conventional devices and the proposed system for 48 h. According to the American Thoracic Society guidelines [42], the 6MWT must be performed indoor along a flat, straight walking course of 30 m and must be supervised by a trained doctor. During the test, the participant must walk on 25-m skidproof mat and move back and forth. In this experiment, smoking participants had been smoking for more than 2 years. The age of participants was higher than or equal to 20 years, and they could walk independently for 6 min. In this study, 15 smoking participants (14 men and 1 woman, mean age: 26.47 ± 2.2 years) and 15 nonsmoking participants (15 males, mean age: 24.53 ± 1.3 years) were recruited.

## 8. Results

### 8.1. Performance of the Proposed Algorithm in Detecting Heart and Breathing Rates

In this section, the performance of the proposed multiparameter analysis algorithm in estimating heart and breathing rates was first evaluated. Figure 7 shows a randomly selected result of detecting breathing events and R waves in ECG. Experimental results show that breathing events and R waves in ECG could be effectively detected using the proposed algorithm. Table 1 and Table 2 show that the binary classification test was used to evaluate the performance of the proposed algorithm, and related parameters were defined as follows: A true positive indicates that the activity event was accurately detected as an activity event; a false positive indicates that nothing was wrongly detected as an activity event; a true negative indicates that nothing was accurately detected as nothing; and a false negative indicates that an activity event was wrongly detected as nothing. Here, a total of 3532 breathing events and 22,742 R waves, which were extracted from 15 nonsmoking people and 15 smoking people during the 6MWT, were used for the test. The sensitivity, positive predictive value, and accuracy were 89.85%, 96.48%, and 87%, respectively, for detecting breathing events and 100%, 100%, and 100 %, respectively, for detecting R waves. The proposed algorithm exhibited an excellent performance in detecting breathing events and R waves and could effectively estimate breathing and heart rates.

### 8.2. Differences Between Multiphysiological Parameters of Different Groups During the 6MWT

In this section, differences between the multiphysiological parameters of smoking and nonsmoking groups during the 6MWT are presented. Figure 8 shows changes in the average breathing rate, heart rate, walking distance, walking speed, and SpO2 value of smoking and nonsmoking groups during the 6MWT. Experimental results show that the breathing rate, heart rate, and walking distance of both smoking and nonsmoking groups increased with time. Here, the breathing rate of the smoking group was significantly higher than that of the nonsmoking group. The heart rate of the smoking group was higher than that of the nonsmoking group. Moreover, after walking for 4 min, the walking distance and speed of the smoking group were significantly lower than those of the nonsmoking group. The SpO2 value of both smoking and nonsmoking groups decreased with time, and the difference between the SpO2 values of different groups was not evident.

## 9. Discussion

Several healthcare monitoring systems [24,25,26] have been developed to monitor cardiovascular and respiratory parameters. A comparison between the proposed system and different monitoring systems is provided in Table 3. In contrast to the aforementioned system, the proposed system could simultaneously monitor real-time ECG, SpO2, and breathing sounds under motion. Moreover, the RFID technique was used in the proposed system. It could effectively provide more precise walking information and avoid variations among different individuals. The proposed multiparameter analysis algorithm could automatically extract heart rate, breathing rate, walking distance, and speed. It exhibited an excellent performance in detecting breathing events and R waves. Finally, the proposed system was successfully applied for monitoring the cardiopulmonary function of smoking and nonsmoking groups during the 6MWT. Experimental results showed that the breathing rate of the smoking group was significantly higher than that of the nonsmoking group. Moreover, compared with the nonsmoking group, the walking distance and speed of the smoking group significantly decreased after walking for 4 min. In general, the exercise capability can be considered a crucial index related to cardiopulmonary function [43]. Therefore, the poor exercise capability of the smoking group could be reflected by the significantly shorter walking distance and speed of the smoking group during the 6MWT. The significantly higher breathing rate of the smoking group may reflect inferior pulmonary function [44]. However, the heart rate of the smoking group was slightly higher than that of the nonsmoking group. Although the SpO2 value of both smoking and nonsmoking groups gradually decreased with time, the difference between the SpO2 values of different groups was negligible. Because no participants had cardiopulmonary disease or anemia, the physical energy consumption of the smoking group during the 6MWT may be insufficient to generate unobvious differences between the heart rate and SpO2 value of different groups.

## 10. Conclusions

In this study, a wearable cardiopulmonary function evaluation system was developed for simultaneous, real-time monitoring of breathing sounds, ECG, and SpO2 during the 6MWT. To effectively measure the indoor walking distance and speed, RFID was used to overcome the concern of parameter variation among different individuals mentioned in previous studies. Moreover, the mechanical design in this system could effectively reduce the influence of motion artifacts on measuring breathing sounds and ECG signals. A multiparameter analysis algorithm was successfully developed to obtain heart rate, breathing rate, walking distance, and speed automatically. Experimental results showed that the proposed system exhibited an excellent performance in estimating these useful physiological parameters, which are related to cardiopulmonary function in the 6MWT. Finally, the proposed system was used to investigate the cardiopulmonary functions of smoking and nonsmoking groups during the 6MWT. Experimental results indicated that the breathing rate, walking distance, and walking speed of the nonsmoking group during the 6MWT were significantly higher than those of the smoking group. Therefore, the proposed system can serve as a more integrated approach for monitoring cardiopulmonary parameters and precise walking information simultaneously during the 6MWT. Moreover, this system can be applied in other clinical or cardiopulmonary researches in the future.

## Figures and Tables

**Figure 1 sensors-19-04656-f001:**
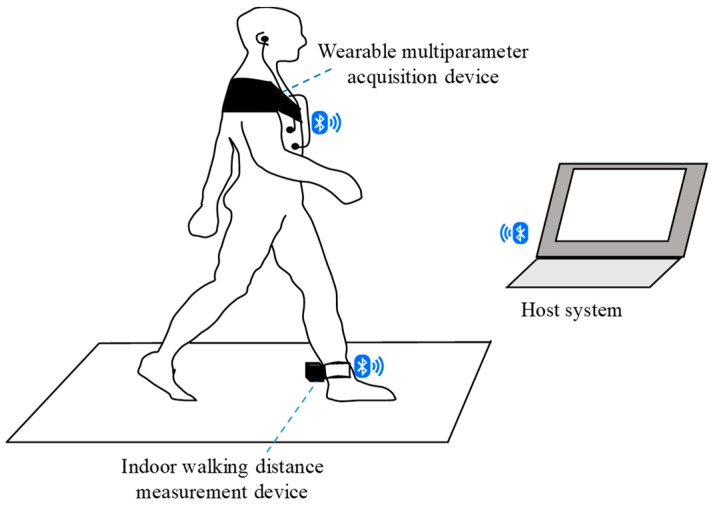
Basic scheme of the wearable cardiopulmonary function evaluation system.

**Figure 2 sensors-19-04656-f002:**
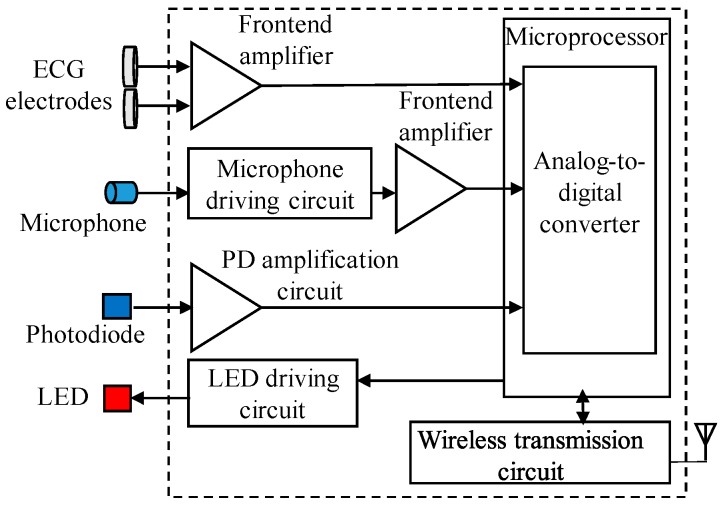
Block diagram of the wireless biosignal acquisition module.

**Figure 3 sensors-19-04656-f003:**
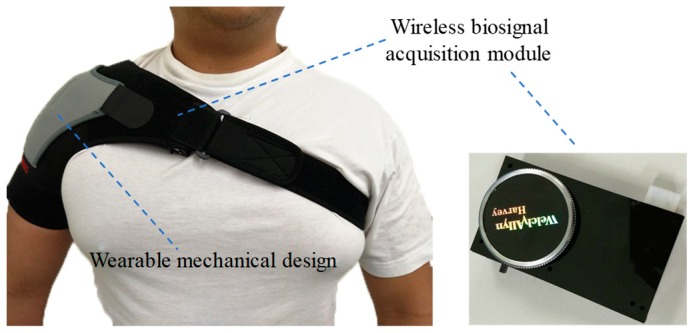
Photograph of the mechanical design in the wearable multiparameter acquisition device.

**Figure 4 sensors-19-04656-f004:**
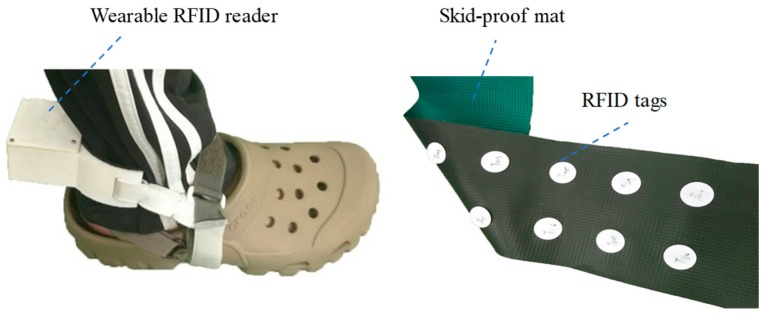
Photograph of the indoor walking distance measurement device.

**Figure 5 sensors-19-04656-f005:**
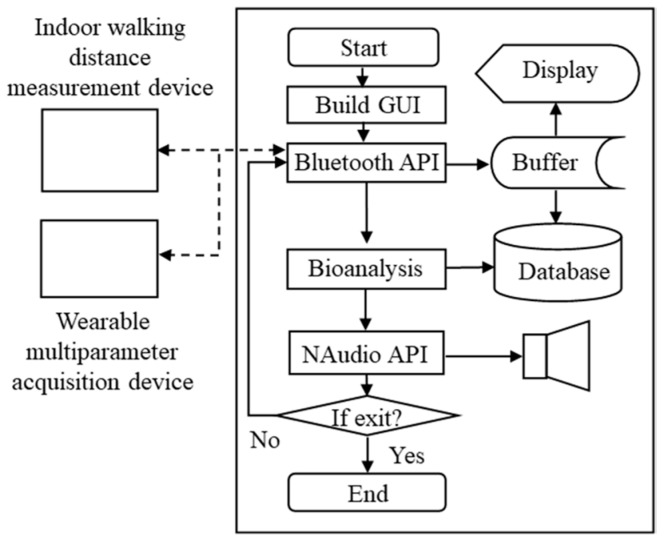
Flowchart of the multiparameter monitoring program.

**Figure 6 sensors-19-04656-f006:**
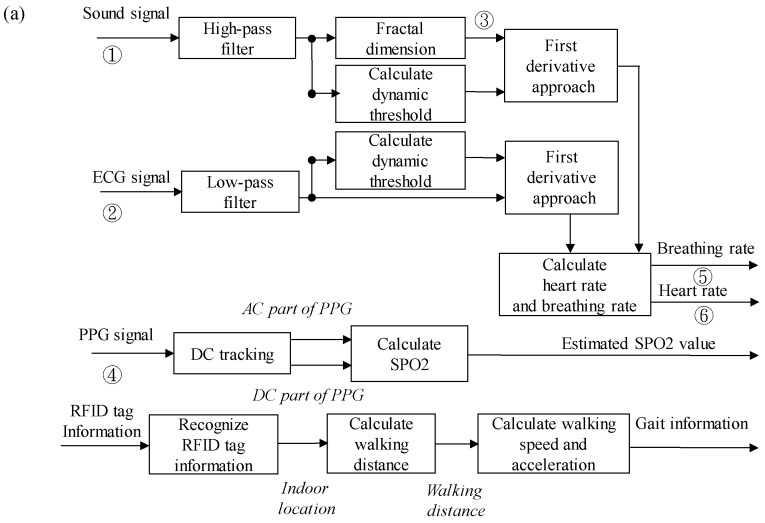
(**a**) Procedure of the multiparameter analysis algorithm. This analysis algorithm processes physiological signals and outputs data, including: (**b**) sound signal, (**c**) ECG signal and R-wave events, (**d**) FD value of sound signal and breathing events, and (**e**) PPG signal.

**Figure 7 sensors-19-04656-f007:**
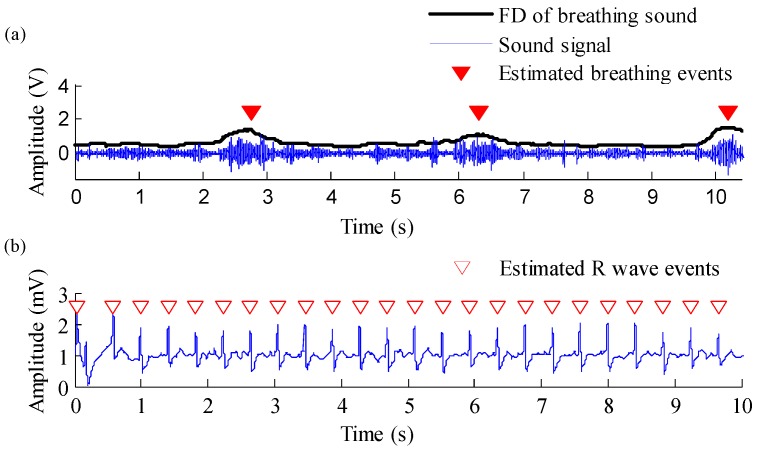
(**a**) Raw signal of breathing sound and estimated breathing events, and (**b**) raw signal of ECG signal and estimated R-wave events.

**Figure 8 sensors-19-04656-f008:**
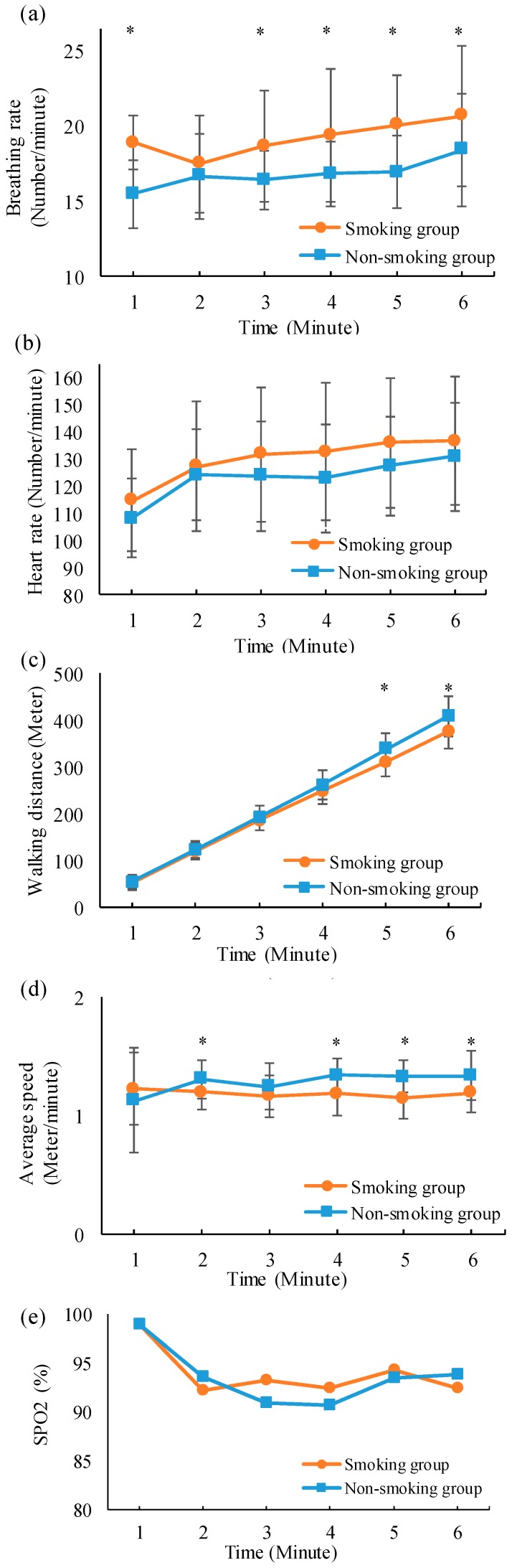
Changes in (**a**) breathing rate, (**b**) heart rate, (**c**) walking distance, (**d**) walking speed, and (**e**) SpO2 between different groups during the 6MWT. Here, * denotes that the difference between smoking and nonsmoking groups is significant (*p* < 0.05).

**Table 1 sensors-19-04656-t001:** Performance of the proposed method in detecting breathing events.

	Estimated Breathing Events			
**Real Breathing Events**		**+**	**–**	**Total**
	**+**	3073 (TP)	347 (TN)	3420
	**–**	112 (FP)	0 (FN)	112
	**Total**	3185	347	3532

**Table 2 sensors-19-04656-t002:** Performance of the proposed method in detecting R-wave events.

	Estimated R Waves			
**Real R Waves**		**+**	**–**	**Total**
	**+**	22742 (TP)	0 (TN)	22742
	**–**	0 (FP)	0 (FN)	0
	**Total**	22742	0	22742

**Table 3 sensors-19-04656-t003:** System comparison between the proposed system and other systems.

	Andreoni et al. [24]	Miramontes et al. [25]	Taffoni et al. [26]	Proposed System
**Physiological Parameters**	ECG, ICG, and Lower-Limb Acceleration	ECG, SpO2, Skin Temperature, Fall Detection, Breathing Rate and Skin Response	Breathing Rate, Heart Rate, Body Movement, Walking Step	SpO2, ECG, Breathing Sound, Breathing Rate, and Walking Information
**Measurement Under Motion**	Yes	No	Yes	Yes
**Sensors**	Triaxial Accelerometer, ECG Electrodes	ECG Electrodes, PPG Sensor, Triaxial Accelerometer, Temperature Sensor, Airflow Sensor, and Galvanic Sensor	PPG Sensor, SDP Sensor, IMU (Accelerometers, Gyroscopes, and Magnetometers)	ECG Electrodes, PPG Sensor, RFID Sensor, and Stethoscope
**Power Supply**	Battery	Battery	Battery	Battery
**Size (cm^3^)**	11.7 × 7 × 2.3	-	-	6 × 6 × 2
**Transmission Mode**	Bluetooth	Wireless Sensor Network	Bluetooth	Bluetooth
**Applications**	6MWT	Cardiopulmonary Function Monitoring	Cardiopulmonary Function Monitoring	6MWT, Cardiopulmonary Function Monitoring
**Disadvantages**	Influence of Motion on Measurement Gait Information	Higher Cost Transmission Architecture, Limited Range of Activities.	Influence of Motion On Heart Rate Estimation	Requirement of RFID Placement
**Monitor Breathing Sound Under Walking**	No	No	No	Yes

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
