# Peer review of "Wearable Cardiopulmonary Function Evaluation System for Six-Minute Walking Test"

_sensors, 2019, doi:10.3390/s19214656_

Round 1

Reviewer 1 Report

I would like to suggest that authors start from the objectives of the cardiopulmonary function test by using 6 mwt, and than review the design of the wearable sensors. We should be able to provide much more information on cardiopulmonary system by using wearable devices than manual test. 

Author Response

Author’s replies to the comments of Reviewer1:

[Comment 1] I would like to suggest that authors start from the objectives of the cardiopulmonary function test by using 6 mwt, and than review the design of the wearable sensors. We should be able to provide much more information on cardiopulmonary system by using wearable devices than manual test.

Answer: Many thanks for the reviewer’s comments. We have reorganized the Section Introduction and added some contents to highlight the objectives of the cardiopulmonary function test by using 6MWT according to reviewer's suggestions.

Reviewer 2 Report

In their work „Wearable Cardiopulmonary Function Evaluation System for Six-minute Walking Test”, the Authors Lin, Jhang and Lin described a novel, wearable system design for cardiopulmonary function evaluation under motion. The presented system comprises ECG, SpO2, respiration sound and gait sensors. The authors demonstrate the estimation of R-wave and breathing event detection using adaptive thresholds, SpO2 estimation, as well as the extraction of gait information including speed and distance. In a 30 subject study they show the influence of smoking on the extracted parameters during a six-minute walking test. The authors conclude that the presented system optimizes gait parameter extraction, is robust to movement artifacts and can be applied in other clinical or cardiopulmonary researches. The study suggested that breathing rate, walking distance and walking speed are significantly higher for the nonsmoking group.

  • L37-38: “However, it still cannot completely represent […]” Please consider revising this sentence. It seems redundant that a people with identical airflow limitation cannot be discriminated based on that parameter.
  • - L62: “A smartphone accelerometer provided nine parameters […]” Please reconsider the term “accelerometer” in this context. Usually, a 9-axial inertia measurement unit (IMU), also oftentimes termed MARG (magnetic, angular rate, gravity) sensor is used here. An accelerometer is but a third of the described systems.
  • - L132-134: Please consider rephrasing the “frequency band” statements. Is the lower corner frequency of a high pass meant? Please clarify. - I recommend an up-to-date literature review regarding wearable auscultation technologies and multimodal, wearable biosignal acquisition.
  • - L166: “[…]was developed by Microsoft C# to monitor […]” Please consider replacing “by” by “in”. - Section 3 (L169-193): Please consider revising and/or shorten this section considerably. The scientific gain from describing the implementation details is at least not immediately clear. Also, textual revision is advices. For example, the term “developed” may have been used in a confusing manner (L182, L185).
  • - L197: The abbreviation “FD” for “fraction dimension” is used before it is introduced.
  • - L201-202: A 50Hz low pass filter for ECG is relatively low. Consider going into detail if that alters the results in any way. - On the ECG Peak detection: the use of a dynamic threshold is oftentimes found in the literature. The well-known Pan-Thompkins algorithms for example uses several thresholds. Please consider detailing if the use of this or a similar algorithms was considered.
  • - In Figure 6. (a) suggests that the “first derivative approach” is a multimodal signal processing step which interlinks the multiple inputs to estimate the breathing- and respiration rate. As detailed in the text, the inputs are processing completely independently. Please consider slightly modifying the schematic overview accordingly.
  • - In equation 2 you state that the calibration of the R-Value to SpO2 is done linearly. While this is sufficient in theory, it may not account for nonlinearities in the system. Please consider giving validation results on the quality of the SpO2 estimation. Please also consider giving details on the calibration procedure.
  • - L274-276: “The proposed algorithm exhibited an excellent performance in detecting breathing events and R waves and could effectively estimate breathing and heart rates.” Please consider moving this into the discussion section. In addition, you did not evaluate the breathing rate and heart rate estimation performance. This could be done for example using Bland-Altman plots or a similar analysis. As this was not performed, a statement should not be made, as the timing of the detection is important for the quality of the results and was also not evaluated.
  • - L303-331: I strongly recommend moving this paragraph to the introduction section L50 and following. This paragraph does not discuss the results of the paper but related work and state of the art.
  • - Considering the amount of related work content, please consider adding a new section after introduction termed “Related work” or similar and move L50 following and L303-331 to that section. This would greatly improve the structure of the paper.

Author Response

Author’s replies to the comments of Reviewer2:

[Comment 1]

L37-38: “However, it still cannot completely represent […]” Please consider revising this sentence. It seems redundant that a people with identical airflow limitation cannot be discriminated based on that parameter.

Answer:

Many thanks for the reviewer’s comments. We have removed the redundant sentences “However, it still cannot completely represent the difference between different people with the same airflow limitation”.

[Comment 2]

L62: “A smartphone accelerometer provided nine parameters […]” Please reconsider the term “accelerometer” in this context. Usually, a 9-axial inertia measurement unit (IMU), also oftentimes termed MARG (magnetic, angular rate, gravity) sensor is used here. An accelerometer is but a third of the described systems.

Answer:

Many thanks for the reviewer’s comments. We have revised the sentence “A smartphone accelerometer provided nine parameters, which were used to estimate gait speed and predict the 6MWD” to “A smartphone accelerometer provided 3-axial acceleration, which were used to estimate gait speed and predict the 6MWD”

[Comment 3]

L132-134: Please consider rephrasing the “frequency band” statements. Is the lower corner frequency of a high pass meant? Please clarify.

Answer:

Many thanks for the reviewer’s comments. The cut-off frequency of the front-end amplifier for ECG was set to 0.1 Hz to remove the lower frequency interference caused by motion. The above description has been added in the revised manuscript.

[Comment 4]

I recommend an up-to-date literature review regarding wearable auscultation technologies and multimodal, wearable biosignal acquisition.

Answer:

Many thanks for the reviewer’s comments. We have collected the complete and latest research regarding wearable auscultation technologies and multimodal, wearable biosignal acquisition to form a new section “Related Work”.

[Comment 5]

L166: “[…]was developed by Microsoft C# to monitor […]” Please consider replacing “by” by “in”.

Answer:

Many thanks for the reviewer’s comments. We have replaced the word “by” by “in” in original Line 166.

[Comment 6]

Section 3 (L169-193): Please consider revising and/or shorten this section considerably. The scientific gain from describing the implementation details is at least not immediately clear. Also, textual revision is advices. For example, the term “developed” may have been used in a confusing manner (L182, L185).

Answer:

Many thanks for the reviewer’s comments. We have shortened the subsection “System Software Design”. We also revised the word “developed” to avoid confusion.

[Comment 7]

L197: The abbreviation “FD” for “fraction dimension” is used before it is introduced.

Answer:

Many thanks for the reviewer’s comments. We have added the complete sentence “fraction dimension” before “FD” in original Line 197.

[Comment 8]

L201-202: A 50Hz low pass filter for ECG is relatively low. Consider going into detail if that alters the results in any way.

Answer:

Many thanks for the reviewer’s comments. As the reviewer’s mention, a 50 Hz low-pass filter for ECG may be relatively low, but it is sufficient for detecting R-waves to estimate heart rate. In the future, we consider using a 150 Hz low-pass filter combined with a 60 Hz notch filter to remove power line interference and reserve most of ECG information.

[Comment 9]

On the ECG Peak detection: the use of a dynamic threshold is oftentimes found in the literature. The well-known Pan-Thompkins algorithms for example uses several thresholds. Please consider detailing if the use of this or a similar algorithms was considered.

Answer:

Many thanks for the reviewer’s comments and suggestion. The proposed algorithm can also provide a good performance on detecting R-wave in ECG. We may test Pan-Thompkins or similar algorithms in the future research.

[Comment 10]

In Figure 6. (a) suggests that the “first derivative approach” is a multimodal signal processing step which interlinks the multiple inputs to estimate the breathing- and respiration rate. As detailed in the text, the inputs are processing completely independently. Please consider slightly modifying the schematic overview accordingly.

Answer:

Many thanks for the reviewer’s comments. According to the reviewer’s comments, Fig. 6 (a) has been modified.

[Comment 11]

In equation 2 you state that the calibration of the R-Value to SpO2 is done linearly. While this is sufficient in theory, it may not account for nonlinearities in the system. Please consider giving validation results on the quality of the SpO2 estimation. Please also consider giving details on the calibration procedure.

Answer:

Many thanks for the reviewer’s comments. In the calibration experiment, the participants were instructed to equip the designed device and a commercial pulse oximeter (DB11, DELBio, Taiwan) simultaneously. Next, these participants would be instructed to hold their breath for about 70 seconds to collect the varying SpO2 data. Finally, a linear regression method was used to obtain coefficients a and b by using R values obtained from the designed device and SpO2 values obtained from the commercial pulse oximeter. The above description has been added in the revised manuscript.

[Comment 12]

L274-276: “The proposed algorithm exhibited an excellent performance in detecting breathing events and R waves and could effectively estimate breathing and heart rates.” Please consider moving this into the discussion section. In addition, you did not evaluate the breathing rate and heart rate estimation performance. This could be done for example using Bland-Altman plots or a similar analysis. As this was not performed, a statement should not be made, as the timing of the detection is important for the quality of the results and was also not evaluated.

Answer:

Many thanks for the reviewer’s comments. The performance of the proposed algorithm on detecting breathing events and R waves have been shown in Table 1 and Table 2. The mentioned sentence has been modified and moved to discussion.

[Comment 13]

L303-331: I strongly recommend moving this paragraph to the introduction section L50 and following. This paragraph does not discuss the results of the paper but related work and state of the art.

Answer:

Many thanks for the reviewer’s comments. We have moved the paragraph (Line 303 to 331) to the end of section “Related Work” (original section “Introduction”).

[Comment 14]

Considering the amount of related work content, please consider adding a new section after introduction termed “Related work” or similar and move L50 following and L303-331 to that section. This would greatly improve the structure of the paper.

Answer:

Many thanks for the reviewer’s comments. We have formed a new section “Related Work” and moved original L303-331 to the end of this section.

Reviewer 3 Report

1) The most important point is that this study involved human trials, but I could not find IRB registration and approval? 

2) In page 4 line 152 to 161, what is advantage of RFID compared with gaitrite? 

3)In page 6 line 208-210, it is not clear about FD calculation. Please use a signal figure to show the L, d, n more straightforward. 

4) In line 232-234, maybe add "as" after "considered"? Also it is confused about the definition of AC and DC. Why is local minimum called DC? Similarly for AC. 

5) In line 246, it is not clear about computation of average walking speed? Two nearest RFID tags every 20 seconds? The distance between two tags is only 9 cm. 

Author Response

Author’s replies to the comments of Reviewer3:

[Comment 1]

The most important point is that this study involved human trials, but I could not find IRB registration and approval?

Answer:

Many thanks for the reviewer’s comments. The institutional review board (IRB 103-3295A3) approved the clinical experiment, Chang Gung Medical Foundation, Taiwan, and the informed consent was signed.

[Comment 2]

In page 4 line 152 to 161, what is advantage of RFID compared with gaitrite?

Answer:

Many thanks for the reviewer’s comments. The GAITRite System is an electronic walkway utilized to measure the temporal (timing) and spatial (two-dimensional geometric position) parameters of its pressure activated sensors. The GAITRite system’s intent is to be utilized as a measuring device for the events occurring during biped and quadruped locomotion. It is a wired, AC-power supported, expensive system.  Our proposed RFID-based walking distance measurement system has wireless, no power, low cost advantages than GAITRite System. The GAITRite System is an independent system so that it is hard to integrate into our whole system. However, the RFID-based walking distance measurement system was developed by ourselves so that it is easy integrated into our whole system.

[Comment 3]

In page 6 line 208-210, it is not clear about FD calculation. Please use a signal figure to show the L, d, n more straightforward.

Answer:

Many thanks for the reviewer’s comments. The definition of FD is illustrated in the below figure. In this example, L = 4, d = 3,  = 1, and n = L/  = 4.

[Comment 4]

In line 232-234, maybe add "as" after "considered"? Also it is confused about the definition of AC and DC. Why is local minimum called DC? Similarly for AC.

Answer:

Many thanks for the reviewer’s comments. According to the reviewer’s comments, we consider using the baseline part and variation part to replace DC and AC parts respectively. The mentioned sentence has been modified in the revised manuscript.

[Comment 5]

In line 246, it is not clear about computation of average walking speed? Two nearest RFID tags every 20 seconds? The distance between two tags is only 9 cm.

Answer:

Many thanks for the reviewer’s comments. First, the diameter of RFID is 3.7 cm. The distance between the centers of every two RFIDs is 9 cm.  In the following figure, if the walker begins to walk from location A, and reaches location B after 20 s. Then we can get the distance is 0.09N m, and the velocity is 0.09N/20 m/s, where N is the number of RFID tags. To avoid a confusion, we have revised to “the averaged walking speed could be estimated from the elapsed time between two locations that is tagged off a nearest RFID tag every 20 s”.

Round 2

Reviewer 1 Report

The work does have certain value in the field.

On other hand, there are some problems with the statements in the article,  for example:

Traditionally, the medical staffs manually collect cardiopulmonary information using different devices. This is not true. Authors do not know the state of the art.  ...and precise indoor gait information (i.e., walking distance, speed, and 17 acceleration). What does "precise indoor gait information" mean? Authors also do not know. Authors really do lack of domain knowledge, which greatly low down the value of the article. 

Author Response

Author’s replies to the comments of Reviewer1:

[Comment 1]

On other hand, there are some problems with the statements in the article, for example:

Traditionally, the medical staffs manually collect cardiopulmonary information using different devices. This is not true. Authors do not know the state of the art.  ...and precise indoor gait information (i.e., walking distance, speed, and 17 acceleration). What does "precise indoor gait information" mean? Authors also do not know. Authors really do lack of domain knowledge, which greatly low down the value of the article.

Answer:

Many thanks for the reviewer’s comments. Actually, the medical staffs in Taiwan, such as Chang Gung Medical Hospital, manually collect cardiopulmonary information using different devices (such as wearable ECG recorder, wearable SpO2 recorder and etc.). About measuring walking distance, the patient walks a 25-m straight floor back and forth, and the medical staff manually counts the walking round trips. To overcome the drawbacks of independence and inconvenience of medical devices. We attempted to propose an integrated and wearable cardiopulmonary function evaluation system for 6MWT. To avoid the confusion, we removed the sentences “Traditionally, the medical staffs manually collect cardiopulmonary information using different devices. These systems are usually big, heavy, non-real-time, and non-wearable design”. In this manuscript, “precise indoor gait information” means patients’ walking position, distance, velocity, and acceleration. To avoid the confusion, we changed the sentence “precise indoor gait information” to “precise walking information”.

Reviewer 2 Report

The authors did improve their work significantly. However, some important improvements should still be considered. - Regarding related work in wearable, multimodal signal acquisition, especially in the context of auscultation and respiratory signal estimation, I recommend a very recent IEEE publication which may fall in line with most of the described requirements: https://ieeexplore.ieee.org/document/8857210 “Wearable Multimodal Stethoscope Patch for Wireless Biosignal Acquisition and Long-Term Auscultation”

  • In Figure 6. (a) suggests that the “first derivative approach” is a multimodal signal processing step which interlinks the multiple inputs to estimate the breathing- and respiration rate. As detailed in the text, the inputs are processed completely independently. Please consider slightly modifying the schematic overview accordingly. The authors stated that Fig. 6a had been changed in this revision, however, no change is apparent.
  • - Throughout the work, gait parameter extraction accuracy is a major concern and considered a major improvement over existing approaches. Even though an accuracy evaluation was not presented, the authors concluded that variations between subjects had been overcome. Please provide data proving that claim.
  • - A major concern was the reduction of motion artifacts. It was claimed that this goal was indeed met, but no data or analysis was shown in that regard. Please provide data proving that claim.

Author Response

Author’s replies to the comments of Reviewer2:

[Comment 1]

Regarding related work in wearable, multimodal signal acquisition, especially in the context of auscultation and respiratory signal estimation, I recommend a very recent IEEE publication which may fall in line with most of the described requirements: https://ieeexplore.ieee.org/document/8857210 “Wearable Multimodal Stethoscope Patch for Wireless Biosignal Acquisition and Long-Term Auscultation”

Answer:

Many thanks for the reviewer’s comments. We have added this reference into section 2.

[Comment 2]

In Figure 6. (a) suggests that the “first derivative approach” is a multimodal signal processing step which interlinks the multiple inputs to estimate the breathing- and respiration rate. As detailed in the text, the inputs are processed completely independently. Please consider slightly modifying the schematic overview accordingly. The authors stated that Fig. 6a had been changed in this revision, however, no change is apparent.

Answer:

Many thanks for the reviewer’s comments. Fig.6 (a) has been modified in the revised manuscript.

[Comment 3]

Throughout the work, gait parameter extraction accuracy is a major concern and considered a major improvement over existing approaches. Even though an accuracy evaluation was not presented, the authors concluded that variations between subjects had been overcome. Please provide data proving that claim.

Answer:

Many thanks for the reviewer’s comments.

The average error of measured walking information is 1.3%. We have added the average error into the subsection 4.3.

[Comment 4]

A major concern was the reduction of motion artifacts. It was claimed that this goal was indeed met, but no data or analysis was shown in that regard. Please provide data proving that claim.

Answer:

Many thanks for the reviewer’s comments. The below figure is the ECG measurement under a sequence of actions (seating, standing, and walking). It shows the influence of motion artifact on ECG measurement can be reduced.

Round 3

Reviewer 1 Report

The modifications are fine now.

Reviewer 2 Report

The authors significantly improved the paper. I have no further objections.